# Higher-Order Spectral Analysis Combined with a Convolution Neural Network for Atrial Fibrillation Detection-Preliminary Study

**DOI:** 10.3390/s24134171

**Published:** 2024-06-27

**Authors:** Barbara Mika, Dariusz Komorowski

**Affiliations:** Faculty of Biomedical Engineering, Department of Medical Informatics and Artificial Intelligence, Silesian University of Technology, Roosevelt 40, 41-800 Zabrze, Poland; dariusz.komorowski@polsl.pl

**Keywords:** atrial fibrillation, ECG, higher-order statistics, bispectrum, CNN, MIT-BIH atrial fibrillation database

## Abstract

The global burden of atrial fibrillation (AFIB) is constantly increasing, and its early detection is still a challenge for public health and motivates researchers to improve methods for automatic AFIB prediction and management. This work proposes higher-order spectra analysis, especially the bispectrum of electrocardiogram (ECG) signals combined with the convolution neural network (CNN) for AFIB detection. Like other biomedical signals, ECG is non-stationary, non-linear, and non-Gaussian in nature, so the spectra of higher-order cumulants, in this case, bispectra, preserve valuable features. The two-dimensional (2D) bispectrum images were applied as input for the two CNN architectures with the output AFIB vs. no-AFIB: the pre-trained modified GoogLeNet and the proposed CNN called AFIB-NET. The MIT-BIH Atrial Fibrillation Database (AFDB) was used to evaluate the performance of the proposed methodology. AFIB-NET detected atrial fibrillation with a sensitivity of 95.3%, a specificity of 93.7%, and an area under the receiver operating characteristic (ROC) of 98.3%, while for GoogLeNet results for sensitivity and specificity were equal to 96.7%, 82%, respectively, and the area under ROC was equal to 96.7%. According to preliminary studies, bispectrum images as input to 2D CNN can be successfully used for AFIB rhythm detection.

## 1. Introduction

Atrial fibrillation (AFIB) is the most common persistent cardiac arrhythmia that poses a challenge in clinical practice and public health [1]. AFIB can lead to severe complications and typically reduce patients’ quality of life and increase mortality as well as the cost to healthcare systems [2]. Scientific reports [3] show that the number of persons with atrial fibrillation in the United States in 2050 will exceed 10 million, while in the European Union, from 2010 to 2060, the number of adults 55 years and over with AFIB will more than double [1,4]. Estimates of current and future incidence and prevalence of atrial fibrillation are worrying and force the constant search for effective screening methods for AFIB detection.

Unlike other arrhythmias, the mechanisms of atrial fibrillation are more complex and have not yet been fully understood [5]. Modern theories link the underlying mechanism of atrial fibrillation with the reentry mechanism, which is not a disorder of impulse generation but its propagation; that is, the movement of an electrical impulse around an abnormal circuit repetitively. The depolarizing wave of the action potential leaves cells in the refractory state, unresponsive to further stimulation; hence, repeated stimulation of the same area requires a time-planned delay so that the next action potential can reenter the circuit and will not encounter cells unable to respond to the stimulation. In anatomic reentry, the boundaries of the circuit are physical cardiac structures [6]. Basic AFIB-maintaining mechanisms can be pointing out local ectopic firing, single-circuit reentry, and multiple-circuit reentry [7]. According to the literature [2], AFIB is definite as *“A supraventricular tachyarrhythmia with uncoordinated atrial electrical activation and consequently ineffective atrial contraction”*.

Pathological features characteristic of AFIB diagnosis based on electrocardiogram documentation include [2,8]: *“lirregularly irregular R-R intervals (when atrioventricular conduction is not impaired), absence of distinct repeating P waves, and irregular atrial activations”*. In compliance with [9], any arrhythmia recorded by a standard 12-lead ECG or a single-lead ECG tracing of at least 30 s of heart rhythm with AFIB characteristics, by convention, is diagnostic of clinical AFIB.

Although AFIB is not a fatal disease in itself if left untreated, it can result in serious health complications and, after some time, may cause stroke, heart failure, heart attack, and premature death [10]. Manual screening of AFIB on electrocardiogram is time-consuming and depends on the experience of the interpreting physician which can lead both to delay and differences in diagnosis and also mistakes and omissions [11]. Artificial intelligence (AI) with machine learning and deep learning techniques using electrocardiography (ECG) are powerful tools for detecting, classifying, and predicting AFIB seem to be a remedy for the above-mentioned human limitations.

Among the deep learning models developed for automatic AFIB detection and classification, we can find deep neural network (DNN) [12,13]; convolution neural network (CNN) [14,15]; recurrent neural network (RNN) [16,17]; long short-term memory network (LSTM) [18,19]; and hybrid (CNN + LSTM) network [20,21,22,23]. Their descriptions with advantages and disadvantages are widely presented and discussed in the work of Murat et al. [20], and Ebrahimi et al. [24]. The main advantage of CNN architectures pointed out in [20] is its strength in obtaining representative properties, so in our studies, CNN with bispectrum input was designed. CNNs can automatically learn representative features directly from the data itself, thus eliminating the need to extract distinctive features manually.

This paper focuses on AFIB detection based on convolution neural networks (CNNs) and higher-order statistics analysis (HOSA). Although many models have been developed using deep learning (DL) [25] to extract information from ECG signals, in this publication, we propose a new approach to detect atrial fibrillation based on higher-order cumulants. In our study, third-order spectra called bispectra are calculated for ECG signals from the MIT-BIH Atrial Fibrillation database and serve as an input to CNN. To our knowledge, such an approach has not yet been described and analyzed in the literature and may contribute to supporting the diagnosis of atrial fibrillation.

This work is organized as follows: in Section 2.1, the theoretical basis for higher-order spectral analysis is presented, and the concept of bispectrum is discussed. Section 2.2 is devoted to the MIT-BIH Atrial Fibrillation database, data preparation, feature extraction, and construction of convolution neural networks used for the AFIB classification and statistic measure. The following Section 3, Section 4 and Section 5 present the results, discussion, and conclusions.

## 2. Materials and Methods

### 2.1. Background on Higher-Order Spectral Analysis

Most biomedical signals present non-linear, non-stationary, and non-Gaussian characteristics [26]. These features cause the widely used primary tools of biomedical signal processing, such as correlation and spectral analysis, to quantify only some of the information available in the biomedical signals. Generally, higher-order spectra are Fourier transforms of higher-order statistics (HOS), moments, or cumulants. Moments are instead used for deterministic signals, while cumulants are for stochastic signals [27].

There are some reasons for using HOS to process biomedical signals to capture the desirable information [26]. Higher-order statistics of the Gaussian process are zero [28], so if the no-Gaussian signal is corrupted by the additive Gaussian noise, the transformation to the higher-order spectra (polyspectra) domain suppresses the Gaussian noise and, in consequence, return the signal with a higher signal-to-noise ratio (SNR) [27]. Because the polyspectra provide high noise immunity, they can be treated as a measure of no-Gaussianity. The Fourier transform, the primary tool of power spectral analysis, transforms the signal from the time domain to the frequency domain. It quantifies the power distribution as a function of frequency but does not provide enough information about the phase relation between harmonic components. Hence, the degree of interaction, that is, phase coupling, usually originating from the non-linear source, cannot be assessed based on the power spectrum [29]. Due to polyspectra’s ability to preserve amplitude and phase information, applying HOSA to process the biomedical signal is natural. Many scientists have applied HOSA with success to various biosignals, such as electroencephalogram (EEG) [30,31], electromyogram (EMG) [32], electrocardiogram (ECG) [33,34,35], heart rate variability (HRV) signals [36], image processing [37], and others.

This paper applied the HOSA technique to reveal hidden information from ECG signals. In particular, the third-order spectrum called the bispectrum was used to capture the impact of atrial fibrillation on ECG and served as an input to CNN.

#### 2.1.1. Moments and Cumulants

Let $X\in R^{N}$ represent the random variable. If the probability density function f(t) of the variable *X* is Gaussian, the random variable *X* is fully characterized by the mean (first-order statistic) and the covariance (second-order statistic). For the continuous random variable *X* with the probability density function f(t) the characteristic function ϕ(t) is defined as (Equation 1)
(1)$$\phi \left(t\right)=E\left(e^{jXt}\right)⟺\phi \left(t\right)=\int_{-\infty }^{\infty }f\left(t\right)\cdot e^{jxt}dt,$$
where E(·) denotes the expectation value, f(t) the probability density function of the random variable *X*. The function ϕ(t) is called the characteristic function or the moment-generating function. The characteristic function ϕ(t) can be expressed as a Maclaurin series expansion (Equation 2)
(2)$$\phi \left(t\right)=1+\sum_{k=1}^{\infty }\frac{m_{k}}{k!}{\left(jt\right)}^{k},$$
with the coefficients $m_{k}=\frac{1}{j^{k}}\phi^{\left(k\right)}\left(0\right)=E\left(X^{k}\right)$ of *k*-order (for k∈{1,2,3,…}) moments of random variable *X* where (*k*) denotes the *k*-order derivative of the function ϕ(t). Higher-order moments are usually used to define the distribution more precisely [38].

Because the higher-order moments of Gaussian distribution are constant but not zero, the natural logarithm of characteristic function is used. The coefficients of expansion of the function g(t)=ln(ϕ(t)), where ϕ(t) is a characteristic function (i.e., moment-generating function), in the Maclaurin series (Equation 3) are known as cumulants ($c_{k}$).
(3)$$g\left(t\right)=\sum_{k=1}^{\infty }\frac{c_{k}}{k!}{\left(jt\right)}^{k},$$
where $c_{k}=\frac{1}{j^{k}}g^{\left(k\right)}\left(0\right)$ is a *k*-order cumulant, and (*k*) denotes the *k*-order derivative of the function g(t). The function g(t) is called a cumulant-generating function. Based on successive derivation of function g(t), the first three cumulants are equal to
$$c_{1}=EX\;\wedge \;c_{2}=E{\left(X-EX\right)}^{2}\;\wedge \;c_{3}=E{\left(X-EX\right)}^{3},$$
that is,
$$c_{1}=m_{1}\;\wedge \;c_{2}=m_{2}-{\left(m_{1}\right)}^{2}\;\wedge \;c_{3}=m_{3}-3m_{2}m_{1}+2{\left(m_{1}\right)}^{3},$$
so the first cumulant is expected value (mean) of random variable *X* and the second and third cumulants are central moments of the second (variance) and third order, respectively. Cumulants on a higher order than two are zero for the Gaussian distribution [28].

Let $X={\left[X_{1},X_{2},X_{3},\ldots ,X_{p}\right]}^{T}$ be p-dimensional random variable $X\in R^{px1}$. The characteristic function of *X* is then definite as (Equation 4)
(4)$$\Phi \left(t\right)=E\left(e^{j(X,t)}\right)⟺\Phi \left(t\right)=E\left(\prod_{k=1}^{p}e^{jX_{k}t_{k}}\right),$$
where $t={\left[t_{1},t_{2},\ldots ,t_{p}\right]}^{T}$, (**t**,X) is a scalar product of vectors X, and **t**. It is said that the vector X admits moments of order n if $E(|X{|}^{n})<\infty$, where $\left|X\right|=\sqrt{\sum_{i=1}^{p}x_{i}^{2}}$ denotes the norm in the $R^{p}$ space. In such a case, the Φ function is n-time differentiable and the *k*-order (k≤n) moments ($M^{\left(k\right)}$) exist and they are definite as (Equation 5) [39]
(5)$$\begin{array}{c} \forall_{(i_{1},\ldots ,i_{k})\in A}\;M^{\left(k\right)}\left(X_{i_{1}},\ldots ,X_{i_{k}}\right)=E\left(X_{i_{1}}\cdot X_{i_{2}}\cdot \cdots \cdot X_{i_{k}}\right)\\ ⇕\\ \forall_{(i_{1},\ldots ,i_{k})\in A}\;M^{\left(k\right)}\left(X_{i_{1}},\ldots ,X_{i_{k}}\right)=\frac{1}{j^{k}}\frac{\partial^{k}\Phi }{\partial t_{i_{1}}\ldots \partial t_{i_{k}}}\left(0\right), \end{array}$$
where the set A contains all the k-element sequences created from the elements of the set {1,2,…,p}. In addition, the function Φ is continuous and Φ(0)=1 with non-zero values in the neighborhood of $0\in R^{p}$ so the function-generating cumulants can be written as (Equation 6)
(6)$$\Psi \left(t\right)=ln(E\left(e^{j(X,t)}\right)).$$

Because of $E(|X{|}^{n})<\infty$, function Ψ is n-time differentiable and the joint cumulants of *k*-order (k⩽n) ($C^{\left(k\right)}$) are definite by (Equation 7)
(7)$$\forall_{(i_{1},\ldots ,i_{k})\in A}\;C^{\left(k\right)}\left(X_{i_{1}},\ldots ,X_{i_{k}}\right)=\frac{1}{j^{k}}\frac{\partial^{k}\Psi }{\partial t_{i_{1}}\ldots \partial t_{i_{k}}}\left(0\right).$$

For multiple dimensional random variables $X\in R^{p}$, ($X={\left[X_{1},X_{2},X_{3},\ldots ,X_{p}\right]}^{T}$) the relationship between its cumulants and moments of *k*-order (k⩽p) could be established by the Leonov and Shiryaev formula [39] (Equation 8)
(8)$$\begin{array}{c} C^{\left(k\right)}\left(X_{i_{1}},\ldots ,X_{i_{k}}\right)=\\ =\sum_{\pi \in Pr(\{1,\ldots ,k\})}{\left(-1\right)}^{|\pi |-1}\left(|\pi |-1\right)!\prod_{B\in \pi }E\left(\prod_{i\in B}X_{i}\right), \end{array}$$
where set Pr({1,…,k}) includes all the blocks of *k*-order partitions, π runs through all blocks of set Pr({1,…,k}), |π| denotes the order (number of parts) for partition in the set π, and B∈π the partition from the set π.

Below, we present an example of Pr({1,2}) for *k* = 2 (Equation 9) and Pr({1,2,3}) for *k* = 3 (Equation 10), where ($|\pi_{i}|=i$ for i∈{1,2,3}).
(9)$$Pr\left(\left\{1,2\right\}\right)=\{\overset{\pi_{1}}{\underset{B_{1}}{\underset{︸}{\overset{︷}{\left\{1,2\right\}}}}}\;;\;\overset{\pi_{2}}{\overset{︷}{\left\{\underset{B_{1}}{\underset{︸}{\left\{1\right\}}},\underset{B_{2}}{\underset{︸}{\left\{2\right\}}}\right\}}}\;\}$$
(10)$$\begin{array}{c} Pr\left(\left\{1,2,3\right\}\right)=\{\underset{B_{1}}{\underset{︸}{\overset{\pi_{1}}{\overset{︷}{\left\{1,2,3\right\}}}}};\\ \overset{\pi_{2}}{\overset{︷}{\left\{\underset{B_{1}}{\underset{︸}{\left\{\{1\},\{2,3\}\right\}}};\underset{B_{2}}{\underset{︸}{\left\{\{2\},\{1,3\}\right\}}};\underset{B_{3}}{\underset{︸}{\left\{\{3\},\{1,2\}\right\}}}\right\}}};\\ \overset{\pi_{3}}{\overset{︷}{\left\{\underset{B_{1}}{\underset{︸}{\left\{1\right\}}},\underset{B_{2}}{\underset{︸}{\left\{2\right\}}},\underset{B_{3}}{\underset{︸}{\left\{3\right\}}}\right\}}}\}. \end{array}$$

Based on (Equation 8)–(Equation 10) for $X=[X_{1},X_{2}]$, the second-order cumulant can be expressed by (Equation 11)
(11)$$C\left(X_{1},X_{2}\right)=E\left(X_{1}X_{2}\right)-E\left(X_{1}\right)E\left(X_{2}\right),$$
while the third-order cumulant is given by (Equation 12)
(12)$$\begin{array}{c} C\left(X_{1},X_{2},X_{3}\right)=E\left(X_{1}X_{2}X_{3}\right)-E\left(X_{1}\right)E\left(X_{2}X_{3}\right)-\\ E\left(X_{2}\right)E\left(X_{1}X_{3}\right)-E\left(X_{3}\right)E\left(X_{1}X_{3}\right)+2E\left(X_{1}\right)E\left(X_{2}\right)E\left(X_{3}\right). \end{array}$$

If we assume that $EX_{i}=0$ for i∈{1,2,3}, then the Formulas (Equation 11) and (Equation 12) simplify to
(13)$$C\left(X_{1},X_{2}\right)=E\left(X_{1}X_{2}\right)\;and\;C\left(X_{1},X_{2},X_{3}\right)=E\left(X_{1}X_{2}X_{3}\right).$$

For time-dependent, real-valued stationary p-dimensional random process $X=[X\left(t\right),X\left(t+\tau_{1}\right),X\left(t+\tau_{2}\right),\ldots ,X\left(t+\tau_{p-1}\right)]$, with the time shift $\tau_{1},\tau_{2},\ldots ,\tau_{p-1}$, respectively. The i-th shift for (i∈{1,2,…,p−1}) is equal to $i\cdot T_{s}$ where $T_{s}$ denotes the sampling period. Hence, the p-order cumulant ($C_{p,X}$) is defined as the p-order joint cumulant of $X\left(t\right),X\left(t+\tau_{1}\right),X\left(t+\tau_{2}\right),\ldots ,X\left(t+\tau_{p-1}\right)$, given by (Equation 14) [40]
(14)$$C_{p,X}=C\left(X\left(t\right),X\left(t+\tau_{1}\right),X\left(t+\tau_{2}\right),\ldots ,X\left(t+\tau_{p-1}\right)\right).$$

The second- and third-order cumulants of zero-mean random variable X can be expressed as follows:(15)$$\begin{array}{c} C_{2,X}\left(\tau \right)=E\left(X\left(t\right)X\left(t+\tau \right)\right)\\ C_{3,X}\left(\tau_{1},\tau_{2}\right)=E\left(X\left(t\right)X\left(t+\tau_{1}\right)X\left(t+\tau_{2}\right)\right). \end{array}$$

If the process X is Gaussian, all the cumulants are zero [28], so the cumulants are also a measure of the deviation from the Gaussian distribution.

#### 2.1.2. Bispectrum

Higher-order spectra are defined with the aid of higher-order moments or cumulants. They are Fourier transforms of higher-order statistics. Applying the Fourier transform for the first three cumulants, we obtain the power spectrum, bispectrum, and trispectrum, respectively [27].

Assuming that the cumulant sequences are absolutely summable (16)
(16)$$\sum_{\tau_{1}=-\infty }^{+\infty }\cdot \cdot \cdot \sum_{\tau_{p-1}=-\infty }^{+\infty }\left|C_{p,X}\left(\tau_{1},\tau_{2},\ldots ,\tau_{p-1}\right)\right|<+\infty$$
the *p*th-order polyspectrum ($S_{p,X}$) is definite as (*p* − 1)-order-dimensional discrete-time Fourier transform (DTFT) of the *p*th-order cumulant as follows (Equation 17):(17)$$\begin{array}{c} S_{p,X}\left(f_{1},f_{2},\ldots ,f_{p-1}\right)=\mathrm{DTFT}\left(C_{p,X}\left(\tau_{1},\tau_{2},\ldots ,\tau_{p-1}\right)\right)=\\ =\sum_{\tau_{1}=-\infty }^{+\infty }\cdot \cdot \cdot \sum_{\tau_{p-1}=-\infty }^{+\infty }C_{p,X}\left(\tau_{1},\tau_{2},\ldots ,\tau_{p-1}\right)\cdot e^{-j\sum_{i=1}^{p-1}2\pi f_{i}\tau_{i}}. \end{array}$$

For *p* = 3, we obtain the bispectrum ($S_{3,X}\left(f_{1},f_{2}\right)=B\left(f_{1},f_{2}\right)$) as the double DTFT of the third-order cumulant $C_{3,X}$ (18)
(18)$$B\left(f_{1},f_{2}\right)=\sum_{\tau_{1}=-\infty }^{+\infty }\sum_{\tau_{2}=-\infty }^{+\infty }C_{3,X}\left(\tau_{1},\tau_{2}\right)\cdot e^{-j\sum_{i=1}^{2}2\pi f_{i}\tau_{i}}.$$

A bispectrum is a function of two frequencies $f_{1},f_{2}$, and their couple $\left(f_{1},f_{2}\right)$ is called a bi-frequency. Because of the symmetry relationships in the cumulants graph, only those bi-frequencies inside the triangular region (19) are considered non-redundant.
(19)$$f_{2}⩾0\;\wedge \;f_{1}⩽f_{2}\;\wedge \;\left(f_{1}+f_{2}\right)⩽\frac{1}{2}F_{s}$$
where $F_{s}$ is the sampling frequency.

### 2.2. AFIB Detection Stage

The proposed method for detecting AFIB rhythms in the ECG signal consists of the following steps: data preparation, feature extraction, and AFIB classification. Feature extraction is based on HOSA, and two architectures of neural networks are proposed for classification: a serial CNN network (further referred to as AFIB-NET) and a pre-trained GoogLeNet network (GNN) [41] with some modifications. The details of each stage are provided in the following subsections. To evaluate the performance of the proposed method, the MIT-BIH Atrial Fibrillation Database (AFDB) (https://physionet.org/content/afdb/1.0.0/ (accessed on 24 June 2024)) was used [42].

#### 2.2.1. MIT-BIH Database

The AFDB database includes 25 long-term ECG recordings of human subjects with atrial fibrillation. Of these, 23 (two of the ECG signals are not provided) records were used in our studies. Each record (of 23) contained two ECG signals sampled at 250 samples per second with 12-bit resolution over a range of ±10 millivolts. Each recording is about 10 h long and has a bandwidth of approximately 0.1 to 40 Hz. AFDB provides the rhythm annotations files, which include 605 annotated episodes, with rhythm annotations of types AFIB (atrial fibrillation), AFL (atrial flutter), J (Nodal (junctional) rhythm), and N used for all other rhythms. AFDB contains a total of 1,686,797.776 (468 h) s of ECG signals (843,398.888 s (234 h) for one ECG lead). The details about time duration for the four available rhythms are included in Table 1.

#### 2.2.2. Data Preparation

In the first stage of data preparation, ECG fragments annotated as AFIB and N (used to indicate all other rhythms) were extracted from each signal of the AFIB MIT-BIH database. Each AFIB and N of the ECG fragment was divided into 5 s segments because it is a long enough time to detect the change in heart rhythm. ECG fragments with AFIB and N rhythms that were less than 5 s long were omitted. As a result of such division, we obtained 134,212 5 s ECG fragments with the AFIB rhythm and, simultaneously, 200,092 5 s fragments with N (all other) rhythms. Fragments of ECG signals with less than 5 s, 582 for AFIB and 574 for N rhythms, were omitted. The bispectrum was then calculated for each 5 s segment of the ECG signals. Bispectra were calculated using the Higher-Order Spectral Analysis (HOSA) MATLAB Toolbox [43]. The HOSA Matlab toolbox proposes two non-parametric methods to estimate the higher-order spectrum (in our case, bispectrum): direct (based on fast Fourier transform (FFT)) and indirect. We checked both methods. The FFT length for computing the bispectrum was set to 512. We chose the direct method for bispectrum estimation based on the preliminary results. The details of algorithms and implementation of direct and indirect methods for bispectrum estimation can be found in the literature [43].

#### 2.2.3. Feature Extraction

In line with a review of the literature in which the frequency range of atrial activities was examined, it is concluded that it is not precisely defined but belongs to the range of approximately 3 to 12 Hz [44,45,46,47,48,49,50]; therefore, for further analysis, we used fragments of bispectra from 0.5 to 12 Hz. The examples of ECG signal and their spectra and bispectra for both AFIB and N rhythms are depicted in Figure 1 and Figure 2, respectively. The absolute value and phase of the bispectrum are presented as a contour obtained for the level equal to 20.

Both the absolute value and the phase of bispectra in the 0.5 to 12 Hz range were normalized into the range between 0 and 1, corresponding to a conversion into a gray image. The min–max normalization method was applied in the area corresponding to the analyzed frequency range. The two-dimensional images representing the absolute value of bispectra size 24 × 24 were used as the input for AFIB-NET. The sample images corresponded to bispectra in the absolute frequency range of 0 to 40 Hz are shown in the Figure 3 and Figure 4. The fragments marked by the red rectangle (after normalization in the zone of interest) were used to train the AFIB-NET network. Figure 5 shows the histograms of pixel value for the normalized data (gray image) corresponding to the amplitude of the bispectrum of the ECG signals with AFIB and N rhythms (for ECG in Figure 1a and Figure 2a).

Since the GNN uses the three color channels (RGB) image with a size of 224 × 224 × 3, the input image preparation was made as follows: we used the fragments of bispectra from 0.5 to 14 Hz (the range of the frequencies was extended to 14 Hz because of GNN input layer size), which corresponds to the image of size 28 × 28 × 1; this image was replicated 64 times (8 × 8), and copied to each of the RGB layers.

#### 2.2.4. Classification

Two different architectures of deep neural networks were used for the classification. The first approach used the series AFIB-NET, whose detailed structure is presented in Table 2, and the second used modified GNN. The modification considered a change in the number of classes and the adaptation of the network output layer for the classification task being performed. The proposed method was evaluated using the bispectrum images corresponding to the ECG signals with AFIB and N rhythms obtained by the methods mentioned above. We considered about 2000 to 18,000 images (half and half AFIB and N class) randomly chosen from all datasets (134,212 images corresponded to AFIB and 200,092 to N (other) rhythms)). The 70% of the obtained bispectra (images) were used to train the AFIB-NET, the 10% was used to validate the network, and the remaining 20% was used to test the networks.

The AFIB-NET was trained for 40 epochs with a mini-batch size (MBS) equal to 32. The initial learning rate (ILR) was calculated using the following formula: ILR=0.1·miniBatchSize/128, resulting in an ILR value equal to 0.25. The GNN was trained for 60 epochs with the initial learning rate equal to 0.0001. The code was written in MATLAB 2022a and ran in the workstation equipment with processor Intel(R) Core(TM) i9-10980XE 3.00 GHz, 128 GB RAM, and graphics card NVIDIA Quadro RTX 5000 with 16 GB memory.

### 2.3. Statistical Measures Applied for the Classifiers Assessment

Several statistical measures were applied for the proposed AFIB-NET and modified GoogLeNet network to evaluate the power of the proposed method dedicated to AFIB detection.

Sensitivity (also known as recall, or true positive rate (TPR)):(20)$$TPR=\frac{TP}{TP+FN}.$$

Specificity (true negative rate (TNR)) understood as:(21)$$TNR=\frac{TN}{TN+FP}.$$

The precision of positive predictive value:(22)$$PPV=\frac{TP}{TP+FP}.$$

The precision of negative predictive value:(23)$$NPV=\frac{TN}{TN+FN}.$$

Prevalence:(24)$$PV=\frac{TP+FN}{TP+TN+FN+FP}.$$

Accuracy
(25)$$ACC=\frac{TP+TN}{TP+FN+FP+TN}=sensitivity\cdot PV+specificity\cdot \left(1-PV\right).$$

The likelihood ratio of positive values:(26)$$LR+=\frac{sensitivity}{1-specificity}$$
and the likelihood ratio of negative values:(27)$$LR-=\frac{1-sensitivity}{specificity}.$$

Area (AUC) under *receiver operating characteristic* (ROC), and F1-score
(28)$$F1=\frac{2\cdot pecision\cdot sensitivity}{pecision+sensitivity},$$
for rhythm classes AFIB ($F1_{AFIB}$, forprecision=PPV) and N ($F1_{N}$, forprecision=NPV). TP—truepositive, FN—falsenegative, FP—falsenegative, TN—truepositive parameters referred to the confusion matrix obtained for the process of classification.

## 3. Results

The statistical measures definite in Section 2.3 were calculated for AFIB-NET and GNN and are summarized in Table 3. The confusion matrices for the proposed AFIB-NET and the modified GNN in the AFIB classification are presented in Figure 6 and Figure 7, respectively. Next to the confusion matrix, we can see vertically (in the blue boxes) the sensitivity and specificity values expressed in percentage. The precision of positive (PPV) and negative (NPV) predictive values are presented in the horizontal blue boxes under the confusion matrix. The values of these parameters are also summarized in Table 3.

In addition, Figure 8 presents ROC curve plots with the mark of the net model operating point and the calculated area under the ROC curve for the AFIB class for both AFIB-NET and GoogLeNet.

## 4. Discussion

### 4.1. Main Findings of the Study

A good classification model is one that minimizes the number of errors (FP and FN in the confusion matrix). However, the cost of these errors is not always the same, that is, not in all applications. From the medical point of view, it is safer to treat a healthy person as sick (FP) and perform further diagnostic tests to exclude the disease than to treat a sick person as healthy (FN) and not undertake any treatment. Therefore, when analyzing both confusion matrices for AFIB detection, the FP>FN situation is acceptable, which, in turn, views lower precision in favor of higher sensitivity. Such a situation occurs for both analyzed networks (Table 3). However, for GNN, the difference is much higher in the benefit of sensitivity, especially for the precision of positive predictive value (PPV).

If we assume that both precision and sensitivity are essential, we can use the F1 metric, which is the harmonic mean of sensitivity and precision and provides information about the balance between sensitivity and precision maintained by the model. A higher F1 score suggests a better balance between sensitivity and precision, indicating a more effective model. After analyzing the results obtained in this context, we found that the AFIB-NET model is more effective than GNN. $F1_{AFIB}$ and $F1_{N}$ for AFIB-NET differ by only 0.007, so the model presents a good high balance. At the same time, GNN is a better-balanced model in terms of sensitivity and precision for class N ($F1_{N}=0.964$) than for class AFIB ($F1_{AFIB}=0.902$), which means that GNN more often classified the person with N rhythm as a person with AFIB, and such feature from a medical point of view can be acceptable.

Based on the sensitivity and specificity measures, we can determine the likelihood ratio of positive and negative predicting values, that is, AFIB or N rhythms detection, respectively. Assuming, in accordance with the literature [53,54], that the results obtained from the model have a real diagnostic value for LR+, about 10 and more, and LR-, about 0.1 and less, the results (LR+ = 15.127 and LR- = 0.050) obtained for AFIB-NET would indicate a greater diagnostic value in relation to GNN. In this case, the results obtained for GNN would classify this model only as applicable.

Sensitivity and specificity are the measures that are the basis for the construction of ROC curves. A good decision rule is one that maximizes both of these measures. We can use ROC curves to find the optimal cut-off model operating point for a given model, that is, the point at which both of these measures reach their maximum value simultaneously. ROC curves illustrate the relationship between the sensitivity and specificity of a given model and allow for a comprehensive assessment of the decision rule of the constructed model [55]. In Figure 8 we can compare the ROC curves for AFIB-NET and GNN. The model operating point for AFIB-NET is closer to point (0,1), which means the maximum values for sensitivity and specificity.

A widely used approach is to calculate the area under the ROC curve (AUC) and treat it not only as a measure of the goodness and rightness of a given model but also as a tool for assessing and comparing classification models. The AUC parameter takes a value in the range [0, 1]. The higher the AUC value, the better the model. For the models considered in this work, the AUC parameter was also determined (Table 3), and its value is equal to 0.983 for AFIB-NET and 0.967 for GNN, so it could be concluded that both models are correct, but the AFIB-NET is slightly better.

The results confirm the potential of the bispectral analysis of ECG signals and the convolution neural networks applied to AFIB detection. One of the problems in automatic ECG analysis is the significant difference in ECG signal morphology between patients or groups of patients and even within the same person [26,56]. For this reason, higher-order spectra, as the spectral representation of higher-order cumulants or moments of the given signal, in our case, bispectrum seems to be a good choice. Bispectrum extracts the no-Gaussian, non-linearity characteristics from ECG signals and reduces the variation in morphology changes. In addition, the bispectrum preserves the phase relationships between harmonic components.

Based on frequency ranges reported to the AFIB rhythm [44,45,46,47,48,49,50], we present the preliminary results of AFIB detection for the bispectrum frequency band 0.5–12 Hz, which are promising, but still, in the future, we would like to check if or how the frequency band influences the analysis’s effectiveness. In the future, we are planning to repeat the proposed methodology for the bispectrum with a higher resolution. In this study, we also tested not only the amplitude bispectrum as the input to the CNN but also the phase bispectrum images and the combined amplitude and phase bispectrum images. However, the obtained results were not satisfactory, and this approach demands further investigation. The conducted learning and testing processes of CNNs were time-consuming and lasted from some to several dozen hours for one set of parameters and one network architecture.

### 4.2. Comparison to Other Algorithms

In line with the summaries and results, which can be found in [57,58,59] and other literature (Table 4), we present a comparison of the classification by the proposed technique with the chosen machine learning, deep learning-based methods, and other algorithms developed for detecting the AFIB rhythm using the MIT-BIH Atrial Fibrillation database. Nevertheless, it should be noted that such a comparison is quite difficult to interpret. A very substantive comparison of AFIB detection results and an interesting discussion on the difficulties and interpretation of comparing the quality of atrial fibrillation detection results for different algorithms are presented in the 2023 Yang et al. [59].

The effectiveness of the proposed procedure for AFIB detection based on the bispectrum images corresponds to those presented in Table 4, in terms of the most often presented measures: sensitivity and specificity. However, the other measures presented in Table 3, discussed earlier are in favor of the bispectrum, which, used as the input to the CNN, has a large potential for automated detection of atrial fibrillation.

### 4.3. Strength and Limitations

This research’s novelty is using bispectrum images as input to CNN to automate AFIB detection. In favor of bispectrum compared to other techniques is bispectrum’s ability to capture the no-Gaussian, non-linear nature of ECG signals. Moreover, bispectrum can identify the phase coupling, usually originating from the non-linear source, and reduce the variation in morphology changes in ECG. The presented preliminary studies confirm that bispectrum images are a good choice for input to the CNN classifier for AFIB detection, but further investigations are required.

The main limitation of this study is that only one ECG database, including two-lead ECG, was used to test the performance of the proposed method. In this work, we tested the ECG signals of the MIT-BIH Atrial Fibrillation database. It is a very good database that is widely used and allows for the comparison of results, but there is a possibility that the signals are somehow correlated. Although the classification was performed with the good practice of using the teaching, validation, and testing set as the next stage of our studies, we also intend to test the proposed methodology on new data from other ECG databases.

## 5. Conclusions

To the best of our knowledge, this is the first study that combines the methodology of higher-order spectrum (bispectrum) and convolution neural network for the automated detection of AFIB in ECG signals. During the classic assessment of the occurrence of AFIB, these three main features *“irregularly irregular R-R intervals (when atrioventricular conduction is not impaired), absence of distinct repeating P waves, and irregular atrial activations”* are taken into account. As the bispectrum suppresses the observed features caused by the morphological changes in ECG, it gives the opportunity to catch the variability contributed by the AFIB. The proposed model uses a bispectrum of ECG signal as an input for the proposed convolution neural network AFIB-NET and detects atrial fibrillation with a sensitivity of 0.953, specificity of 0.937, and AUC equal to 0.983. The obtained results for AFIB-NET confirm the effectiveness of the use of bispectra in combination with CNN for detecting AFiB. As the future direction of our studies, we plan to test other bands of frequency and resolution for bispectra images and to test the proposed method for additional ECG databases. We also intend to extend the AFIB-NET classifier for more ECG rhythms.

## Figures and Tables

**Figure 1 sensors-24-04171-f001:**
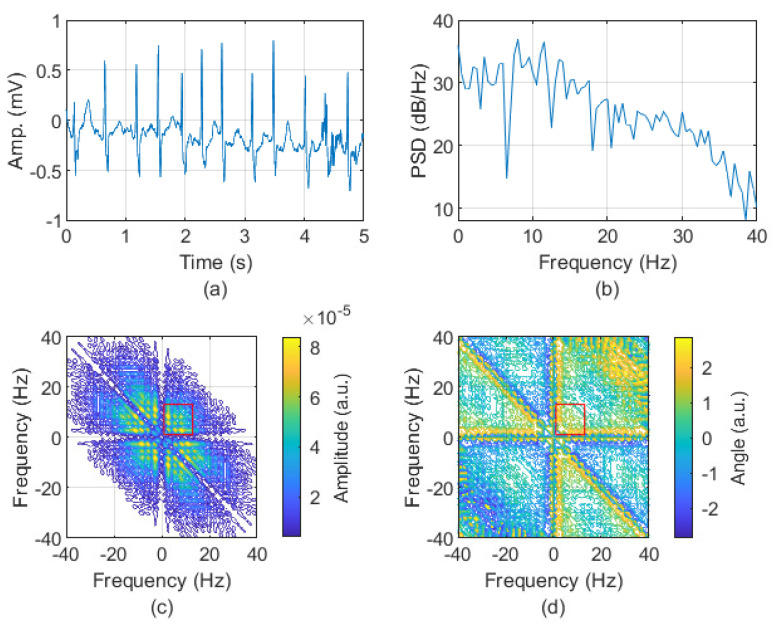
The 5 s length fragment of ECG with AFIB rhythms (signal: 04048 AFDB, channel 1, the first 5 s of the signal containing AFIB rhythm) (**a**). The spectrum (calculated by Welch method [51,52]) (**b**). The absolute value (**c**) and phase (**d**) of bispectrum. The red rectangle indicates the frequencies in the range 0.5 to 12 Hz.

**Figure 2 sensors-24-04171-f002:**
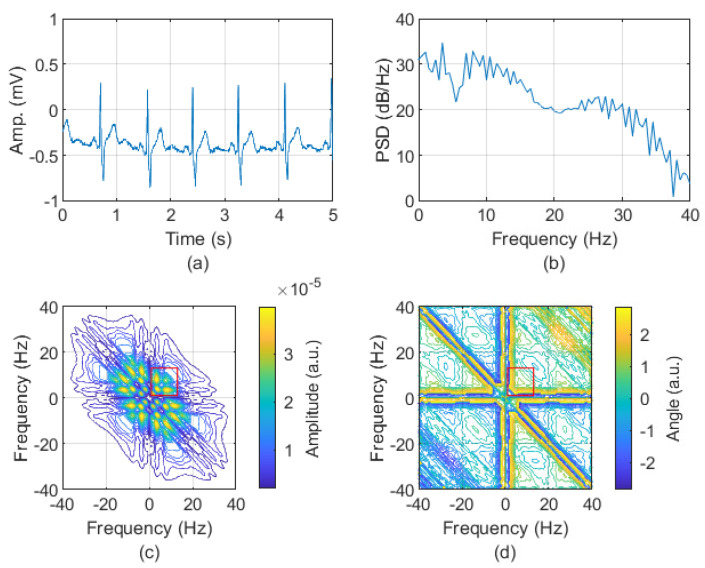
The 5 s length fragment of ECG with N rhythms (signal: 04048 AFDB, channel 1, the first 5 s of the signal containing N rhythm) (**a**). The spectrum (calculated by Welch method [51,52]) (**b**). The absolute value (**c**) and phase (**d**) of bispectrum. The red rectangle indicates the frequencies in the range 0.5 to 12 Hz.

**Figure 3 sensors-24-04171-f003:**
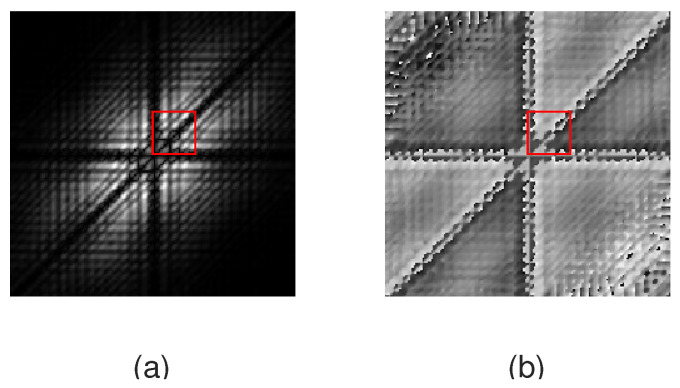
The gray images (absolute value (**a**) and phase (**b**)) obtained after converting the bispectrum of the fragment of ECG with the AFIB rhythm. The red rectangle (**a**) marks the area used as the input of the AFIB-NET. The size of images corresponds to the absolute frequency range of 0 to 40 Hz (signal: 04048 AFDB, channel 1, the first 5 s of the signal containing AFIB rhythm).

**Figure 4 sensors-24-04171-f004:**
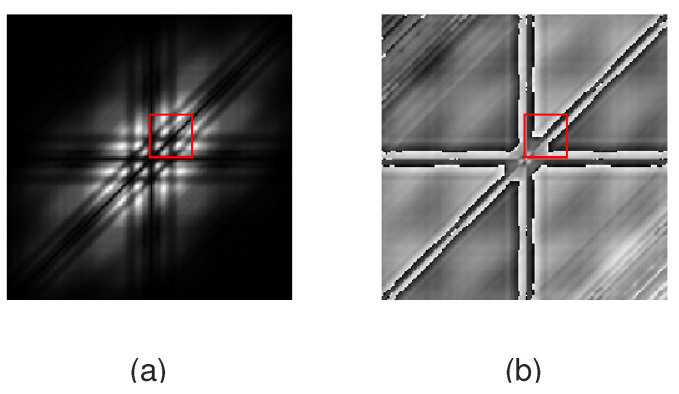
The gray images (absolute value (**a**) and phase (**b**)) obtained after converting the bispectrum of the fragment of ECG with N rhythm. The red rectangle (**a**) marks the area used as the input of the AFIB-NET. The size of images corresponds to the absolute frequency range of 0 to 40 Hz (signal: 04048 AFDB, channel 1, the first 5 s of the signal containing N rhythm).

**Figure 5 sensors-24-04171-f005:**
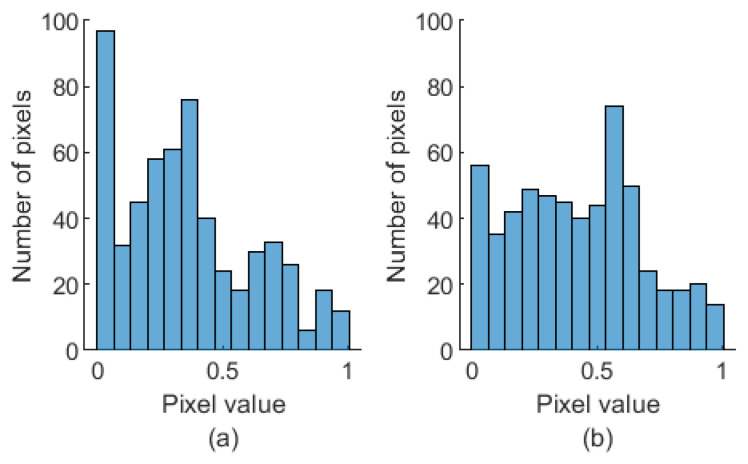
The histograms of the images correspond to the absolute value of the bispectrum of the ECG signal with the AFIB rhythm (**a**) (signal: 04048 AFDB, channel 1, the first 5 s of the signal containing AFIB rhythm) and the N rhythm (**b**) (signal: 04048 AFDB, channel 1, the first 5 s of the signal containing N rhythm), respectively.

**Figure 6 sensors-24-04171-f006:**
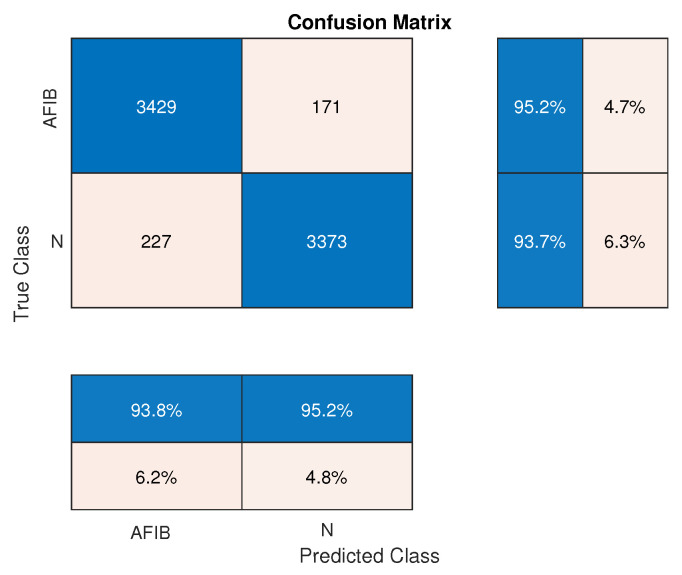
Confusion matrix for the proposed AFIB-NET in AFIB classification. The vertical blue boxes next to the confusion matrix present the sensitivity (TPR) and specificity (TNR) values, while the horizontal blue boxes below the confusion matrix state the precision of the positive (PPV) and negative (NPV) predictive values, respectively.

**Figure 7 sensors-24-04171-f007:**
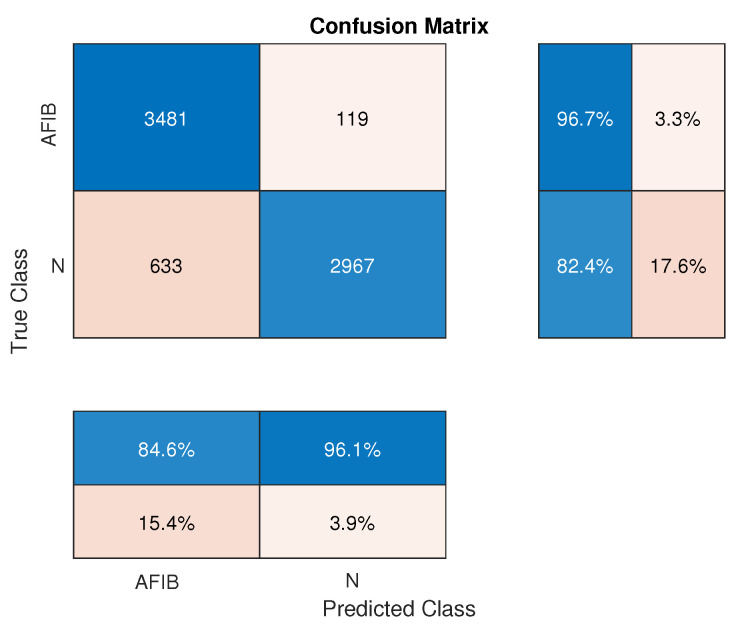
Confusion matrix for modified GNN in AFIB classification. The vertical blue boxes next to the confusion matrix present the sensitivity (TPR) and specificity (TNR) values, while the horizontal blue boxes below the confusion matrix state the precision of the positive (PPV) and negative (NPV) predictive values, respectively.

**Figure 8 sensors-24-04171-f008:**
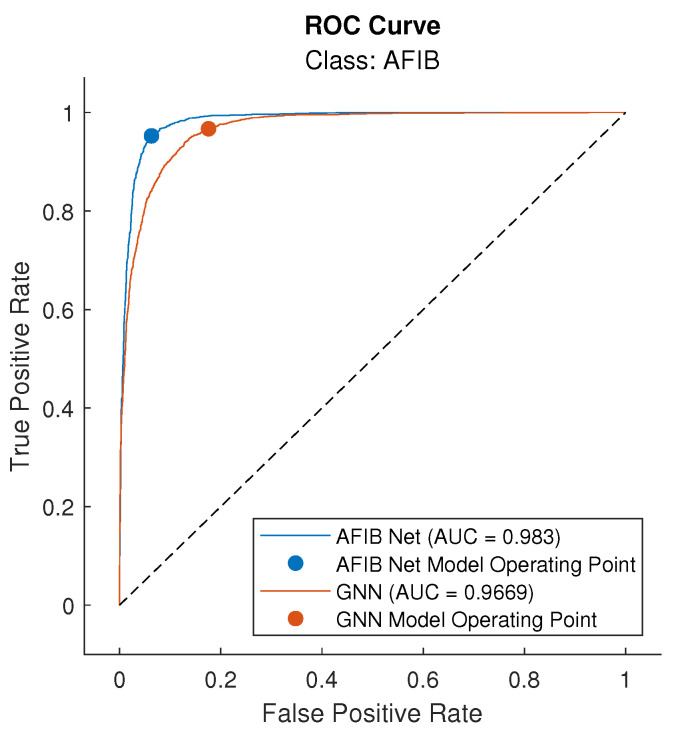
Receiver operating characteristic (ROC) curves of AFIB class for AFIB-NET and GNN, with marked net model operating points and areas under ROC (AUC) value. *True Positive Rate* means sensitivity, while *False Positive Rate* states 1-specificity.

**Table 1 sensors-24-04171-t001:** Time duration of the AFIB, AFL, Nodal, N (other) rhythms for analyzed ECG signals from MIT-BIH Atrial Fibrillation database.

Episode	№	Min	Max	Mean	Total
		[s]	[s]	[s]	[s]
AFIB	291	1.684	36,822.864	1155.436	336,231.984
AFL	14	3.532	3390.912	419.794	5877.116
Nodal	12	1.524	86.016	27.582	330.980
N (other)	288	4.252	30,981.372	1739.440	500,958.808

**Table 2 sensors-24-04171-t002:** The AFIB-NET neural network architecture. The activation dimension (Activations) is given in the following format: S × S × C × B, where labels mean: S-spatial, C-channel, and B-batch observation.

Layer	Type	Filter Size	Number of Filters	Stride	Activations	Number of Learnables
1	Image Input	-	-	-	24 × 24 × 1 × 1	0
2	Convolution	3 × 3	8	[1 1]	24 × 24 × 8 × 1	80
3	Batch Normalization	-	-	-	24 × 24 × 8 × 1	16
4	ReLu	-	-	-	24 × 24 × 8 × 1	0
5	Max Pooling	-	-	[1 1]	23 × 23 × 8 × 1	0
6	Convolution	3 × 3	16	[1 1]	23 × 23 × 16 × 1	1168
7	Batch Normalization	-	-	-	23 × 23 × 16 × 1	32
8	ReLu	-	-	-	23 × 23 × 16 × 1	0
9	Ma × Pooling	-	-	[1 1]	22 × 22 × 16 × 1	0
10	Convolution	3 × 3	32	[1 1]	22 × 22 × 32 × 1	4640
11	Batch Normalization	-	-	-	22 × 22 × 32 × 1	64
12	ReLu	-	-	-	22 × 22 × 32 × 1	0
13	Max Pooling	-	-	[1 1]	11 × 11 × 32 × 1	0
14	Convolution	3 × 3	64	[2 2]	11 × 11 × 64 × 1	18,496
15	Batch Normalization	-	-	-	11 × 11 × 64 × 1	128
16	ReLu	-	-	-	11 × 11 × 64 × 1	0
17	Max Pooling	-	-	[1 1]	5 × 5 × 64 × 1	0
18	Convolution	3 × 3	128	[2 2]	5 × 5 × 128 × 1	73,856
19	Batch Normalization	-	-	-	5 × 5 × 128 × 1	256
20	ReLu	-	-	-	5 × 5 × 128 × 1	0
21	Fully Connected	-	-	-	1 × 1 × 2 × 1	6402
22	Softmax	-	-	-	1 × 1 × 2 × 1	0
23	Classification Output	-	-	-	1 × 1 × 2 × 1	0

**Table 3 sensors-24-04171-t003:** Classifier evaluation parameters: sensitivity, specificity, positive predictive value (PPV), negative predictive value (NPV), prevalence (PV), accuracy (ACC), the likelihood ratio of positive value (LR+), the likelihood ratio of negative value (LR-), the area under ROC curve (AUC), and F-score for AFIB and N class for both the AFIB-NET and GNN.

Measure	AFIB-NET	GooLeNet (GNN)
Sensitivity (TPR)	0.953	0.967
Specificity (TNR)	0.937	0.824
PPV	0.938	0.846
NPV	0.952	0.961
PV	0.500	0.500
ACC	0.945	0.896
LR+	15.127	5.494
LR-	0.050	0.040
AUC	0.983	0.967
$F1_{AFIB}$	0.945	0.902
$F1_{N}$	0.952	0.964

**Table 4 sensors-24-04171-t004:** Comparison of sensitivity and specificity of chosen deep learning-based methods and other algorithms for AFIB rhythm detection for ECG signals using the MIT-BIH Atrial Fibrillation database.

Method Proposed by	Sensitivity (%)	Specificity (%)
RADHAKRISHNAN et al. (2021) [58]	99.17	98.90
Marsanova et al. (2020) [60]	96.32	98.61
Wang et al. (2020) [61]	98.70	98.90
Mousavi et al. (2019) [57]	99.53	99.26
Xia et al. (2018) (STFT) [14]	98.34	98.24
Xia et al. (2018) (SWT) [14]	98.79	97.87
Kumar et al. (2018) [62]	95.80	97.60
Tripathy et al. (2017) [63]	97.77	98.67
Asgari et al. (2015) [64]	97.00	97.10
Lee et al. (2013) (RMSSD) [65]	90.49	94.17
Lee et al. (2013) (ShE) [65]	74.15	96.81
Lee et al. (2013) (SamE) [65]	97.26	99.61
Huang et al. (2011) [66]	96.10	98.10
Tateno et al. (2001) [67]	94.40	97.20
AFIB (Proposed work)	95.30	96.70
GNN (Proposed work)	93.70	82.00

## Data Availability

To evaluate the performance of the proposed method, the MIT-BIH Atrial Fibrillation Database was used [42] https://physionet.org/content/afdb/1.0.0/ (accessed on 24 June 2024).

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
