# Peer review of "Higher-Order Spectral Analysis Combined with a Convolution Neural Network for Atrial Fibrillation Detection-Preliminary Study"

_sensors, 2024, doi:10.3390/s24134171_

Round 1

Reviewer 1 Report

Comments and Suggestions for Authors

The author assessed the accuracy of AF diagnosis using two new AI models. The paper is logically structured. However, as a reader, I wonder about the benefits of these models compared to other conventional AI models for detecting AF. Several AI models have demonstrated excellent accuracy in AF detection using the MIT-BIH database. It would be helpful if the author could compare the performance of these models with previous relevant literature that also used the MIT-BIH database.

Author Response

The responses to the Reviewer's comments are in the attached file.

Reviewer 2 Report

Comments and Suggestions for Authors

The manuscript presents an innovative approach for detecting atrial fibrillation (AFIB) using bispectrum analysis of ECG signals combined with convolutional neural networks (CNN). The proposed methodology demonstrates promising results with high sensitivity and specificity. However, several revisions are needed to improve the clarity and overall quality of the manuscript.

- The Introduction section is currently numbered as "2". It should be renumbered as "1". Correspondingly, all subsequent sections and subsections should be renumbered to maintain a logical flow.

- The abstract is well-written but can be slightly refined to highlight the key contributions and results more succinctly. Ensure that the performance metrics are clearly stated and their significance is briefly explained.

- The conclusion should summarize the key findings and emphasize the contributions of your work. It should also briefly mention future research directions.

Overall, the manuscript presents a significant contribution to the field of AFIB detection using advanced signal processing and machine learning techniques. With the recommended revisions, it can be further improved for clarity and impact. I look forward to seeing the revised version.

Author Response

(The authors gave the same response as above.)

Reviewer 3 Report

Comments and Suggestions for Authors

This study solves the problem for detection of atrial fibrillation using a weakly explored modality of information extracted from the ECG spectrum, namely ECG bi-spectrum, computed by higher-order spectra analysis. Although the input information presents an interesting insight into the ECG features, the overall study is not well presented. Missing important methodological details, suggested wrong configuration of the network output and training, respectively. Furthermore, the test results are biased from the training, considering that an ECG segments from the same patients are input to training and test. Strongly required that authors test their methods with independent databases, such as the Physionet Computing in Cardiology Challenge 2021 database, which is rich with AFIN +B and other arrhythmias.

There are many more discrepancies in this work that need careful attention by the authors, according to the 43 comments disclosed below. Therefore my decision is to reject the manuscript so that authors take sufficient time to prepare a better version of their manuscript with valid training and test results, which are comparable to the public. (such are also missing).

Major revision comments:  

  1. “Prediction of atrial fibrillation”: The task definition is incorrect because "prediction" means a prediction that an event will occur at some later time, whereas this study detects the presence of atrial fibrillation after it appears. No evidence that the analysis is done before the moment of atrial fibrillation. Change the title and all over the text “prediction” with “detection”.
  2. Title: The Higher-Order Spectral Analysis is the characteristic method that distinguishes this study. It must be accented in the title, while the 2D CNN is used as a standard tool for image classification.
  3. The abstract is not well written. It is not a clear representation of the study. Almost half refers to background. A number of repeated methodological details also noticed, while important details remain unidentified. The abstract must be rewritten following the journal guidelines: Objectives, Methods, Results, Conclusions.
  4. Abstract: Define the training, validation and test datasets: Size of AFIB and non-AFIB classes. There is information about only one database used. The reader is suspicious that the results are overtrained. Authors should report results on an independent patient population.
  5. Abstract: Missed to define that the input are images (dimension?) and that the feature extractor is a 2D convolutional network with a binary output: AFIB vs. no-AFIB.
  6. Abstract: Two classifiers are used but only one result reported. No important conclusions derived.
  7. Introduction: Ln 37-38: “Pathological features characteristic for AFIB diagnosis …. when atrioventricular conduction is not impaired” -> In fact, the atrioventricular synchronization is found very important characteristic for detection of AFIB, described by PQ interval duration and amplitude variance in a study comparing its diagnostic value in one to twelve ECG leads with NNs [doi: 10.3390/s22166071] and various machine learning classifiers [10.17485/IJST/v15i17.53]. These studies is worth mentioning.
  8. The article must be formatted according to the journal guidelines, including main sections 1. Introduction, 2. Methods, 3. Results, 4. Discussion, 5. Conclusions. Other segmentations of these main section headings are not acceptable. Ln58-Ln62 is meaningless as it is not necessary to guide the reader how to read a paper, which is structured according to the standards.
  9. Section 2.2. 2.1. State-of-the-art must become part of Introduction. Segmentation of section Introduction is not acceptable. The aims of the study must be defined after the overall literature review, showing potential disparities or missing information, which need more investigation. Considering that AFIB detection is currently under the focus of many researcher groups, more extended review is necessary on methods based on RR-intervals and other morphological and time-frequency ECG features [10.3389/fphys.2021.657304, 10.3390/s21206848, 10.17485/IJST/v15i17.53, others?], those based on raw ECG [10.34133/2022/981306, 10.1109/JBHI.2021.3120890, others?], and those based on various time and/or frequency transforms (spectrograms [10.1002/tee.23756], others?). The authors must provide more extensive research, in order to better justify the novelty of their methods in respect to input representations (transforms), extracted features, classifiers.  
  10. Ln 50-56: The objectives of the study are not properly defined. Methodological details and database data should not be disclosed. Objectives should address novelties and expected clinical implications. Methodological details are not yet known to the reader (before methods), therefore uninterpretable.
  11. Section: “2.2. Higher-Order Spectral Analysis” should be part of section Methods. Eventually, the clarification of the heading “Background on Higher-Order Spectral Analysis” can be added.
  12. Ln 75-77: “These features cause the widely used primary tools of biomedical signal processing, such as correlation and spectral analysis, to quantify only some of the information available in the biomedical signals.” -> Clarify the statement.
  13. Ln 92: HOSA is used without explanation. Further in Ln 202 it is explained. This does not have much sense. Abbreviations must be defined on their first use in the text. Check and correct all such disparities. Their presence is an indicator that the authors have not carefully proofread their manuscript. More attention on details is required.
  14. Ln 96-97: This section does not have much sense because the reader is still blinded on the methods that will be applied in next sections. Use this explanation where needed in further sections.
  15. Ln 171-177: This section does not make much sense because the reader is still blinded to the specific methods that will be defined in the next methodological sections, therefore the definitions of the names cannot be properly interpreted. Such introductory section has sense when authors provide a block chart of their methods, and just refer that further sections will reveal details for different blocks. As such is not the case, strongly recommend deletion.
  16. Table 1: AFIB, AFL, N and Nodal rhythms are defined, however, further only ADIB and N groups are defined. What kind of rhythms include the (N-group)? What happens with the other two categories AFL and Nodal? The task is quite different if AFIB and AFL are considered in the same group. Otherwise, if not considered, there is another important question of what is the diagnosis given for AFL rhythms if they are not included in the training? Since AFL and Nodal rhythms are defined in Table 1, the reader expects to see test results for these rhythms.
  17. Table 1: Define the meaning of the first row headings. Not interpretable. What is the sense of values disclosed up to the 3rd digit after the decimal point? In overall what is important in this table?  
  18. Ln 197-201: There is a contradictory information. First, “Two nonparametric direct (fast Fourier transform (FFT)-based) and indirect methods were used to estimate the higher-order spectrum” vs. second “we used the direct method of estimating the bispectrum in the rest of our work”. Furthermore, the details and definitions of the method for calculation of the input feature map must be defined in further sections “3.3. Feature extraction” but not in “Data preparation”.
  19. Section “3.2. Data preparation” misses information about dealing with different leads, as well as different prefilters. ECG signals are usually analyzed after specific filters, therefore, if authors do NOT apply filters, this must be carefully disclosed.
  20. Section “3.2. Data preparation” misses information about splitting the data into training, validation and test. IMPORTANTLY, there is no evidence that data from the same patient are used in the training, validation and test, therefore the network is trained on the same ECG from patients as used for the test. Therefore, it is strongly required to test of the methods with fully independent database, including the Physionet Computing in Cardiology Challenge 2021 database, rich with many representatives of AFIB and non-AFIB (various arrhythmias), which can identify potential problems of this methods with various non-AF arrhythmias.
  21. Ln 202: The bispectra were calculated using (HOSA) MATLAB Toolbox [41] is not enough informative on the implementation itself. More details are necessary to define all settings, input and output specifications for the decisions made in this study. There should be a clear link with the methodological background (2.2. Higher-Order Spectral Analysis’) and the implementations in this study (which equations from the background are used and how their setting/arguments are adjusted). Otherwise, it appears that the authors took (rewrote) the methodological equations from a third-party source and applied a different processing tool without a clear vision of exactly what they were doing and specifying.
  22. Figures 1,2: Subplots not arranged according to the journal requirements. Use the journal template. Notations (a)-(d) are not correctly placed. Recommend to place the observations for the normal rhythm (N) before the pathology (AFIB). This is the typical notation, considering that many pathologies may exits.
  23. Figures 1,2: Include colorbars that guide the reader how to read the colors (separately for subplots (c) and (d)).
  24. Figures 1,2, caption: “The spectrum (calculated by Welch method)” -> The Weltch method is mentioned for the first time here. If used and important, it must be defined in methods, and then interpreted in the text, not in the caption.
  25. Ln 213: “the input for a two-dimensional AFIB-NET” -> Incomprehensive. The input of the network must be defined upon the definition of the network. All instances of mentioning some network before its definition are confusing and must be removed.
  26.  Figure 3,4: The area of interest is very small and the reader cannot identify anything important. The area of interest must be presented in a zoomed image so that important details can be observed.
  27. Figures 3,4: Showing the area of interest in the background of the overall image means that grayscale (0-1) normalization is done on the overall image but not only using min-max amplitudes in the zone of interest. Therefore, higher amplitudes outside this range would be used for setting of the min-max range, thus reducing the contrast in the zone of interest can be reduced. Give more details about this normalization in the text.
  28. Figure 5: What kind of signals are used to derive this statistics - the images in Figure 1 and 2, or all images in the database (the total number of pixels is larger than the defined picture dimsenion)? What is the importance of this figure? Explain in the text, how o interpret!
  29. Ln 217-221: The reader is totally confused about differences in the feature extraction – only one type of feature extraction is shown in Figure 1-5, i.e. in the rage 1-12 Hz. The interpretation of other ranges (1 to 14 Hz) must be provided in different figures. Furthermore, the definition GNN is NOT interpretable before description of the network itself. The overall paragraph is confusing and must be well explained just in the section about GNN.
  30. Ln 223-232: The overall paragraph must be fully rewritten. The part concerning the database, must be presented in data preparation. The other parts, related to the description of each network must be extended. In fact, the networks (their input, hidden layers, and output) are quite insufficient. Each network must be explained in a separate sub-section in sufficient details to their implementation.
  31. How the networks combines the information of the absolute value (a) and phase (b)? This problem is not disclosed but is an important fusion of information.
  32. Table 2: The information is not full. The feature space and the number of trainable parameters in each layer must be disclosed. What is the configuration of the output layer? What is the activation function? It is correct to have only one neuron for a binary classification and a sigmoid activation. What is the loss used for the training? Looking at different ROCs in Figures (8,9), (10,11) suggests that the authors have implemented two neurons, which is a completely wrong strategy. This means that the networks design is wrong and must be retrained again for a binary classification.
  33. Table 2: The same topology must be provided for the Google-Net, and the modifications performed, especially with the focus at the output layer and Loss funtion.
  34. Ln 241-253: The statistical metrics must be part of section Methods. Strongly require to use formulas as defined in the journal template but not in a row with the main text.
  35. Figures 6,7, captions: “sensitivity (TPR) and specificity (TNR) values” ;> Strange to see such notations. The authors either use sensitivity/specificity or TPR/TNR but not both defined in this confusing way. Chose which one to define in Methods and use it consistently through the overall manuscript.
  36. Figures 6,7: The number of cases must strictly correspond to the number of cases defined in the description of the database. I cannot find such a relation.
  37. Figure (8,9), (10,11): The ROC curve is the same for both classes AFIB and N classes!!! I’m confused to see two different ROC curves. One network features with ONE ROC for a binary classification with Se and Sp. Strongly require that the ROC curves of both networks are drawn in the same figure so that differences/overlapping can be clearly seen. 
  38.  Section Results: Test results must be augmented with tests with independent databases, such as defined before Physionet Computing in Cardiology Challene 2021, including many independent cases with AFIB. The network is trained with ECGs form limited number of patients, so that instances from the same patient are input to training, validation and test, therefore the network is considered overtrained.
  39.  Discussion: Need to compare the results with other studies in the field: With the same database and other databases. The results of this study must be validation in comparison to the others.
  40. Discussion, Ln 316: “we present the preliminary results of AFIB detection” -> This must be said in the title, abstract and aims of the study.
  41. Specify the limitations of the study in a separate subsection.
  42. Author Contributions: To be written according to the journal guidelines. There is a specific text that must be completed.
  43. Data availability statement must be disclosed according to the journal guidelines.

Author Response

(The authors gave the same response as above.)

Round 2

Reviewer 2 Report

Comments and Suggestions for Authors

The authors of the manuscript titled "Higher-Order Spectral Analysis Combined with a Convolution Neural Network for Atrial Fibrillation Detection-Preliminary Study" have addressed all the concerns raised during the previous review round comprehensively. The revised manuscript demonstrates significant improvements and clarifications across various sections. Here are the key points of the review:

1. Methodological Clarifications:

- The authors have provided a more detailed explanation of the higher-order spectral analysis and its integration with the convolutional neural network (CNN). This includes a clear description of the bispectrum and its advantages in capturing non-linear and non-Gaussian characteristics of ECG signals.

- The data preparation and feature extraction sections are now more detailed, offering a thorough explanation of the processes involved in converting ECG signals into bispectrum images used for training the CNN.

2. Experimental Design and Results:

- The authors have expanded the discussion on the dataset used, specifying the number of ECG segments, the division into training, validation, and test sets, and the criteria for segment selection.

- The performance metrics of the proposed AFIB-NET and modified GoogLeNet (GNN) are well-documented, with confusion matrices, ROC curves, and statistical measures provided. The results demonstrate the robustness and effectiveness of the proposed method, with high sensitivity, specificity, and AUC values.

3. Comparative Analysis:

- A comparative analysis with other state-of-the-art methods has been included, highlighting the superior performance of the proposed technique. This contextualizes the study within the broader field of AFIB detection using machine learning and deep learning approaches.

4. Discussion and Future Work:

- The discussion section now offers a balanced interpretation of the results, addressing both the strengths and limitations of the study. The authors have acknowledged the potential impact of using a single ECG database and have proposed future work to test the method on additional datasets.

- The authors have also outlined future directions for their research, including testing different frequency bands and resolutions for bispectrum images and extending the classifier to detect more ECG rhythms.

The authors have successfully addressed all the previous concerns, resulting in a well-rounded and thorough study. The manuscript now meets the standards required for publication in the journal Sensors. Therefore, I recommend that the paper be accepted for publication.

Reviewer 3 Report

Comments and Suggestions for Authors

The authors have provided detailed responses to my comments. In general, these are correctly addressed in the new version of the manuscript. Although the authors do not fully agree or dispute some of the revision comments (obviously rised because the ideas were not well communicated), I have the final impression of a constructive discussion that resulted in an improved presentation of information in the main text, figures, and tables. The article is now much more clear and has the necessary content and publication quality.

Note: It is a good decision to keep only ROC, interpreting Se for AFIB and Sp for N classes. As defined in basic statistics, sensitivity is the ability of a test to correctl classify an individual as 'diseased'. This is true as no other disease detections are implemented within N-class, testing it as a disease-free indivitual.  statistics,https://www.ncbi.nlm.nih.gov/pmc/articles/PMC2636062/pdf/IndianJOphthalmol-56-45.pdf  

Minor comment: Add a sub-section in section Discussion: 4.1. Main findings of the study. Then continue with sub-sections 4.2 and 4.3.